# Impact of Double-Stranded RNA Internalization on Hematopoietic Progenitors and Krebs-2 Cells and Mechanism

**DOI:** 10.3390/ijms24054858

**Published:** 2023-03-02

**Authors:** Genrikh S. Ritter, Anastasia S. Proskurina, Maria I. Meschaninova, Ekaterina A. Potter, Daria D. Petrova, Vera S. Ruzanova, Evgeniya V. Dolgova, Svetlana S. Kirikovich, Evgeniy V. Levites, Yaroslav R. Efremov, Valeriy P. Nikolin, Nelly A. Popova, Aliya G. Venyaminova, Oleg S. Taranov, Alexandr A. Ostanin, Elena R. Chernykh, Nikolay A. Kolchanov, Sergey S. Bogachev

**Affiliations:** 1Institute of Cytology and Genetics of the Siberian Branch of the Russian Academy of Sciences, 630090 Novosibirsk, Russia; 2Institute of Chemical Biology and Fundamental Medicine of the Siberian Branch of the Russian Academy of Sciences, 630090 Novosibirsk, Russia; 3Department of Natural Sciences, Novosibirsk National Research State University, 630090 Novosibirsk, Russia; 4State Research Center of Virology and Biotechnology “Vector”, Novosibirsk Region, 630559 Koltsovo, Russia; 5Research Institute of Fundamental and Clinical Immunology, 630099 Novosibirsk, Russia

**Keywords:** dsRNA, internalization, hematopoietic progenitors, Krebs-2 cancer stem cells, stimulation of colony growth

## Abstract

It is well-established that double-stranded RNA (dsRNA) exhibits noticeable radioprotective and radiotherapeutic effects. The experiments conducted in this study directly demonstrated that dsRNA was delivered into the cell in its native form and that it induced hematopoietic progenitor proliferation. The 68 bp synthetic dsRNA labeled with 6-carboxyfluorescein (FAM) was internalized into mouse hematopoietic progenitors, c-Kit+ (a marker of long-term hematopoietic stem cells) cells and CD34+ (a marker of short-term hematopoietic stem cells and multipotent progenitors) cells. Treating bone marrow cells with dsRNA stimulated the growth of colonies, mainly cells of the granulocyte–macrophage lineage. A total of 0.8% of Krebs-2 cells internalized FAM-dsRNA and were simultaneously CD34+ cells. dsRNA in its native state was delivered into the cell, where it was present without any signs of processing. dsRNA binding to a cell was independent of cell charge. dsRNA internalization was related to the receptor-mediated process that requires energy from ATP. Synthetic dsRNA did not degrade in the bloodstream for at least 2 h. Hematopoietic precursors that had captured dsRNA reinfused into the bloodstream and populated the bone marrow and spleen. This study, for the first time, directly proved that synthetic dsRNA is internalized into a eukaryotic cell via a natural mechanism.

## 1. Introduction

Gamma radiation is short-wavelength electromagnetic radiation resulting from the discharge of nuclei in an excited state, during radioactive decay and in nuclear reactions. The risk of atomic cataclysms and the damaging effects of γ radiation have intensified the search for adequate protection from this type of ionizing radiation. Other types of ionizing rays have less energy, and physical methods of protection can be used as a defense.

The effect of ionizing radiation on a living organism damages its functional systems and causes death. Ionizing radiation is believed to have the strongest impact on membrane structures and the cell nucleus. Membrane lysis causes cell structure disruption, while defects in the nuclear DNA disturb the total functional integrity of chromatin and cause abnormal cell division, chromosomal aberrations, and apoptosis [1]. Poorly differentiated bone marrow cells, testicular germ cells, and the intestinal and skin epithelium are the main target organs for γ rays [2,3]. The radiosensitivity of the entire mammalian organism is considered equivalent to that of hematopoietic cells since their aplasia resulting from whole-body irradiation with the lowest absolute lethal dose causes the death of the organism.

A radioprotective effect is understood as a reduction in the frequency and severity of post-radiation damage to biomolecules and (or) stimulation of the processes of their radiation repair.

The most efficient radioprotectors are two classes of chemical compounds [4,5]: (i) sulfur-containing radioprotectors (aminothioles) acting as molecular traps of free radicals and (ii) indolylalkylamine derivatives (agonists of biologically active amines, which can induce acute hypoxia and metabolic suppression through specific cellular receptors) [6,7,8].

As mentioned earlier, ionizing radiation exhibits the most detrimental effect on the DNA molecule in the nuclear chromatin. Chromosomal damage, together with the subsequent aberrant mitosis and cell death, is another mechanism of cytoreductive activity of ionizing radiation. Chromatin DNA exposure to active metabolites causes damage to these molecules already described in the literature, of which the double-strand breaks are the most lethal kind. If the cellular repair mechanisms of such DNA damage are disrupted, apoptosis is initiated.

A previous study [9] described the novel principle of radioprotective action that neither relates to direct protection against γ-quantum nor limits the exposure to oxidative stress induced by secondary radicals but is rather characterized by the successful post-irradiation recovery of hematopoietic stem cell precursors associated with the involvement of dsRNA fragments in repair. The external “corrector” added to the repair process is eventually responsible for the recovery of the hematopoietic and immune systems and the preservation of the viability of organisms exposed to irradiation.

As noted, dsRNA exhibits noticeable radioprotective effects. The key characteristics of dsRNA molecules that define their ability to protect experimental animals against lethal doses of γ radiation have been identified [9,10]. Bone marrow cells of γ-irradiated mice exhibited the classic pattern of double-stranded breaks when assessed by comet assay, antibodies to γH2X histone, and production of specific repair proteins. Results showed that 160–200 µg dsRNA per mouse injected one hour before radiation exposure protects the experimental animals against the lethal radiation dose. The radioprotective efficacy was identical to or higher than that of the conventional radioprotector B-190 (Russia). Analysis of the blood cell morphology after sublethal irradiation (8 Gy, LD60/30) showed that dsRNA injections preserve the viability of progenitors and accelerate the recovery process, which is associated with the correct repair of nuclear structures. A prolonged radioprotective effect of dsRNA was described when it was found that 160–200 µg dsRNA per mouse injected on day 4 post-irradiation prevented the death of 100% of the experimental mice. The survival of the experimental mice correlated with the emergence of a large number of proliferating leukocytes in the spleen, forming typical lymphoid cell colonies. dsRNA was not degraded in mouse plasma; its size remained unchanged after 2 h of incubation. The double-stranded form of RNA, rather than its sequence, is vital for radioprotective activity. A designed synthetic dsRNA probe, FAM-dsRNA, was detected in CD34+ hematopoietic multipotent progenitors after 1 h of incubation with bone marrow cells. This result suggests that the animals’ protection from lethal doses of radiation is associated with the presence of dsRNA molecules in hematopoietic progenitors. At the same time, the mobilization of progenitors to the periphery after irradiation and treatment with a dsRNA and the development of splenic colonies confirm the hypothesis that hematopoietic progenitors saved from γ radiation are mainly involved in the radioprotective effect. Radioprotection is associated with the preservation and rapid recovery of leukocytic and erythroid lineages. It was determined that the protective effect does not correlate with repair proceeding via the non-homologous end-joining mechanism but with repair occurring via the homologous recombination mechanism induced by radiation-triggered damage in the cells [9,10].

A hypothesis has been put forward proposing that dsRNA is preserved in peripheral blood and delivered into hematopoietic stem cells (c-Kit+, a marker of long-term hematopoietic stem cells) or poorly differentiated bone marrow cells (CD34+, a marker of short-term hematopoietic stem cells and multipotent progenitors), where it becomes involved in repair through the homologous recombination mechanism [11]. Although the details of its involvement remain unclear, the process ensures proper restoration of the chromatin continuum in hematopoietic progenitors. Hematopoietic progenitors are rescued and remain viable. Viable progenitors become mobilized and are anchored to the splenic parenchyma and bone marrow, where they proliferate. As a result, the hematopoietic and immune systems are recovered.

This elegant, logical scheme of the involvement of dsRNA molecules in the recovery of hematopoietic stem cells or multipotent progenitors after γ-radiation-induced damage contains three events lacking unambiguous experimental verification. The first event is the delivery of dsRNA molecules into the cell in its native form. The second is the induction of hematopoietic stem cells or multipotent progenitor proliferation by dsRNA. The third event is homing, and re-fixation of hematopoietic progenitors mobilized from the bone marrow into the bloodstream, in the bone marrow and spleen of mice.

The experiments conducted in this study directly demonstrated that dsRNA is delivered into the cell in its native form and that this dsRNA induces the proliferation of hematopoietic stem cells or multipotent progenitors. It was also demonstrated that the bone marrow and spleen are critical organs for hematopoietic stem cells or multipotent progenitors in the bloodstream of healthy animals.

## 2. Results

### 2.1. CD34+ Poorly Differentiated Bone Marrow Cells Are Capable of Internalizing FAM-dsRNA. Treating Bone Marrow Cells with Synthetic dsRNA Stimulates Colony Formation by Hematopoietic Precursors

Using FAM-dsRNA, we showed that cells internalizing the labeled material were detected in a bone marrow specimen and were surrounded by several stromal cells arranged into a rosette-shaped cluster (Figure 1a). The morphology of the rosettes, which retained their integrity after bone marrow resuspension, may indicate that the FAM+ cell belongs to and resides in the center of the hematopoietic stem cell bone marrow niche. Treating the bone marrow with synthetic dsRNA induced the formation of hematopoietic precursor cell colonies. Cells of the granulocyte–macrophage lineage were most sensitive to treatment (Figure 1b, Table 1). Bone marrow cell typing indicated that both hematopoietic stem cells and poorly differentiated bone marrow cells (c-Kit, CD34+) were simultaneously FAM+ cells (Figure 1c,d). This result suggested that mouse bone marrow progenitors internalized the fragments of extracellular dsRNA.

Hematopoietic progenitors as a system of cells are only 0.01–0.1% or less of all bone marrow cells. This factor did not allow us to use this model to assess the internalization of dsRNA fragments into intracellular compartments of hematopoietic progenitors. To study the dsRNA internalization into a eukaryotic cell with subsequent extrapolation of the obtained data to hematopoietic progenitors, we used the ascitic form of mouse Krebs-2 carcinoma as a working model system.

### 2.2. Comparative Analysis of Internalization of FAM-dsRNA and TAMRA-dsDNA Fragments into CD34+ Krebs-2 Cells

Our studies have used the standardized dsDNA probe, the PCR-labeled TAMRA+ human *AluI* fragment, for many years. It was shown that TAMRA-dsDNA is internalized by a small (0.1–3.0%) population of poorly differentiated cells belonging to different types of cellular communities (mesenchymal stem cells, c-Kit/CD34+ hematopoietic progenitors, and tumor cells of different lineages, including the Krebs-2 mouse ascites carcinoma [12,13,14]). It was also revealed that TAMRA+ Krebs-2 cells simultaneously carry the surface marker CD34 of poorly differentiated bone marrow cells. Additionally, 95% of TAMRA+ cells are simultaneous CD34+ cells, while only half of CD34+ cells can internalize TAMRA-dsDNA [12].

The first part of this study, as well as Ritter et al. [9] detected the FAM-dsRNA in CD34+ poorly differentiated bone marrow cells. An analysis of CD34+/FAM+ stem cells revealed them to be only 0.01–0.1% of all bone marrow cells. This fact limited the possible analysis of the internalization mechanism of dsRNA fragments into CD34+ hematopoietic progenitors using bone marrow cells as the model system.

Considering the above, an assumption was made that CD34+ Krebs-2 cells are capable of internalizing dsRNA fragments along with their ability to internalize dsDNA fragments. A series of experiments was performed, and the findings showed that CD34+ Krebs-2 cells could internalize FAM-dsRNA. The primary analysis demonstrated that FAM fluorochrome is not internalized by Krebs-2 ascites cells. Furthermore, it was previously shown that fluorochrome-labeled triphosphate also does not internalize into Krebs-2 cells [12]. Only the fluorescently labeled polymeric form of dsRNA is detected in Krebs-2 ascites cells (Figure 2a).

It has repeatedly been demonstrated that various Krebs-2 ascites specimens contained between 0.1% and 11% of FAM-dsRNA positive or TAMRA-dsDNA positive cells. The comparative data on the ability of Krebs-2 cells to internalize FAM-dsRNA or TAMRA-dsDNA obtained in three particular experiments are reported below.

When TAMRA-dsDNA was used, Krebs-2 ascites contained 0.3% CD34+ cells, 8.1% TAMRA+ cells, and 0.3% CD34+/TAMRA+ cells (Figure 2(b1)). When FAM-dsRNA was used, the distribution in Krebs-2 ascites was 1.7% CD34+ cells, 1.8% FAM+ cells, and 0.8% CD34+/FAM+ cells (Figure 2(b2)).

Experiments were performed to analyze the colocalization of TAMRA-dsDNA and FAM-dsRNA in Krebs-2 ascites cells. It was found that 2.1% of cells were double-positive for both markers. Meanwhile, 2.0% of ascites cells internalized FAM-dsRNA exclusively, while 0.7% of cells internalized exclusively TAMRA-dsDNA (Figure 2(b3)). This finding suggested general patterns of internalization of both nucleic acid types into Krebs-2 ascites cells, and it is possible that there is also an independent pathway through which nucleic acids penetrate the intracellular compartments of cells that show a potency for internalization.

In order to shed light on the interplay between TAMRA-dsDNA and FAM-dsRNA in their interaction with Krebs-2 cells and outline avenues for further research, we conducted experiments on the cross-competitive inhibition of the internalization of labeled TAMRA-dsDNA and FAM-dsRNA by identical unlabeled PCR fragments of dsDNA and the synthetic dsRNA duplex. The results of these experiments were difficult to interpret but generally indicated several alternative internalization routes whose mechanisms are currently being studied.

We estimated the duration of FAM-dsRNA internalization by the cell. Confocal microscopy data showed that the cell becomes saturated with dsRNA molecules within 40 min, and then the saturation curve reaches a plateau (Figure 2c,d).

### 2.3. Direct Evidence for the Internalization of Synthetic dsRNA into a Eukaryotic Cell (Exemplified by Krebs-2 Ascites Cells)

Our present and previous study [9] employed a synthetic 68 bp dsRNA that comprises two parts related to mouse ribosomal RNA and was labeled during 6-FAM synthesis at the 3′ end (Figure 3a).

Confocal analysis of FAM-dsRNA localization in the Krebs-2 cell was performed in the first step. The images in Figure 3b show that the fluorochrome-labeled material is localized in the nucleus of the Krebs-2 cell.

Numerous attempts employing various approaches, including stem-loop PCR, did not allow us to directly demonstrate that dsRNA is internalized by the cell. This was because only a small amount of material was available for the study, and there was an obvious lack of specificity. The alternative experimental approach was developed and implemented to directly prove that dsRNA has been internalized. Synthetic dsRNA was labeled with both FAM and radioactive phosphorus at its 5′ end. Krebs-2 ascites cells were incubated in the presence of FAM-γP^32^-dsRNA. The cells were then sorted into FAM+ and FAM− cells. The sorted cells were evaluated according to whether or not they incorporated radioactive phosphorus. No radioactivity was detected in FAM− cells. After sorting and measuring radioactivity, RNA was extracted from the cells, and polyacrylamide gel electrophoresis was performed. The results indicate that the Krebs-2 cell (eukaryotic cell) contains intact FAM-γP^32^-dsRNA (Figure 3c,d). Analyzing the mechanism of interaction between nucleic acids and the eukaryotic cell [15], we hypothesized that a robust chemical or molecular interaction could exist between nucleic acid fragments and glycocalyx factors. In order to directly confirm the localization of the labeled material in the nucleus, nuclei were isolated from the sorted FAM+ cells. For the overall population of sorted cells in the precipitate, the count rate was 30 counts per second (a particular experiment). The count rate for the isolated nuclei in the precipitate was 25 counts per second (the same experiment). The nuclei were subjected to freeze–thaw lysis. This treatment allowed for determining the degree of dsRNA integration with the nuclear factors (Figure 3e,f). Direct analysis of the incorporation of radioactive phosphorus into the whole cell and the nucleus, as well as radiation counting data, showed that dsRNA is almost entirely contained in the nucleus and is associated with nuclear factors.

### 2.4. Direct Evidence for the Internalization of Synthetic dsRNA into a Eukaryotic Cell (Exemplified by Krebs-2 Ascites Cells)

#### 2.4.1. Determining the Modes of Interaction between FAM-dsRNA and Cell Surface Elements

The following actions were taken to identify the modes of interaction between FAM-dsRNA and cell surface elements (physical trapping/positive cell surface charge): (a) a series of free-flow electrophoresis experiments were performed, and the percentage of FAM+ cells at the opposite poles of the electrophoresis chamber was assessed, (b) the charge of the cells internalizing a double-stranded RNA probe was evaluated using a high-specificity positively charged dye, and (c) the charge of cells internalizing FAM-dsRNA was assessed using heparin to block the positive charge.

##### Free-Flow Gel Electrophoresis of Krebs-2 Cells Followed by Quantification of Cells Internalizing FAM-dsRNA at the Opposite Poles of the Electrophoresis Chamber

A method for determining the cell charge by free-flow electrophoresis in saline solution was developed, as other electrophoretic systems (TAE, TBE, or FB) caused degenerative changes in cell morphology and their osmotic degradation. The essence of the method was as follows. The cells were placed into a dialysis bag, hermetically sealed on both sides. “Bridges” made of Whatman 3 MM paper were placed on both sides of the bag, and electrophoresis was performed for no longer than 40 min. Partial multidirectional migration of differently charged cells occurred. The experiments with red blood cells and TAMRA-dsDNA showed that red blood cells migrate towards the positive pole, while TAMRA+ cells migrate towards the negative pole [15]. After the electrophoresis, Krebs-2 cells were taken out of the respective side of the dialysis bag and treated with FAM-dsRNA, and the number of FAM+ cells was estimated either microscopically or by flow cytometry. In total, seven experiments were performed. Unlike with TAMRA-dsDNA [15], the percentage of cells internalizing FAM-dsRNA at both poles did not differ significantly (Figure 4a,b). This finding could imply that the primary contact between dsRNA and the cell, to a great extent, depends on factors other than the cell surface charge, as it has been reliably shown for dsDNA [15].

##### Assessing the Mode of Internalization of FAM-dsRNA Using the High-Specificity Positively Charged Basic Blue 41 Dye

In order to confirm the first results obtained by determining the charge of FAM+ cells, we performed experiments to neutralize the charge of Krebs-2 cells using Basic Blue 41 dye (cationic or positively charged dye) and subsequently estimated the percentage of FAM+ cells (Figure 4c–e). If internalization is dependent on cell surface charge and unrelated to any other mechanism present on the negatively charged cells, then varying the order in which FAM-dsRNA and Basic Blue 41 are added will not change the pattern of distribution of the labeled material, and the cells will reside either outside or partially outside the cloud of cells stained with the cationic dye. Otherwise, there will be visible changes in the distribution of cells labeled with fluorochrome and cationic dye. In this case, cloud overlap can be attributed to the fact that there is an alternative mechanism of interaction between dsRNA and the cell.

FAM+/Basic Blue 41 results. Some FAM+ cells lay outside the cloud of cells stained with cationic dye. This fact may indicate that, in this case, the internalization of negatively charged dsRNA molecules occurred in positively charged cells. The partial cloud overlap can be explained by the fact that some cells carrying a positive charge contain the positively charged membrane components, which are responsible for the internalization of a certain amount of dsRNA.

Basic Blue 41/FAM+ results. The distribution pattern was completely changed. Almost all FAM+ cells resided in a cloud of cells also labeled with cationic dye. We interpret these results as suggesting at least two vectors of association between dsRNA and Krebs-2 cells due to the charge, and an alternative mechanism of interaction with the cell membrane. Notably, the second mechanism responsible for the link is charge-independent and predominant among these two mechanisms. Thus, blocking the negative charge using the cationic dye opens this pathway for the cells whose membrane is negatively charged, so dsRNA interacts only via this mechanism.

##### Assessing the Charge of dsRNA-Internalizing Cells Using Heparin

Our studies have found that heparin completely inhibits the internalization of TAMRA-dsDNA into Krebs-2 cells [16]. These findings implied that both dsDNA and heparin compete for cell surface-binding elements. Experiments to evaluate the internalization of FAM-dsRNA were performed. Heparin, at the maximum experimental concentration (4 U), does not inhibit the internalization of dsRNA (Figure 4f,g). This fact may imply that the association with dsRNA molecules proceeds through an alternative interaction mechanism independent of the heparin-mediated pathway, unlike that for dsDNA molecules.

#### 2.4.2. Characterizing the Changes in the Spatial Position of FAM-dsRNA Associated with the Small Population of Krebs-2 Cells after Microgel Electrophoresis

The experimental design based on microgel electrophoresis of cells was elaborated. It was expected that, depending on the method used for immobilizing FAM-dsRNA on the cell membrane, electrophoresis would visualize changes in the positions of dsRNA fragments on the cell membrane (as shown previously for dsDNA fragments [15]). The following parameters were chosen. Cells were incubated in the presence of the probe for 5 and up to 60 min. Electrophoresis was performed either involving two 15 min steps (cells visualized at each step) or for 30 min without any intermediate evaluation. The following morphological variants of changes in the pattern of interaction between FAM-dsRNA and Krebs-2 cells were revealed. For most FAM-dsRNA-labeled cells, the position of the labeled material in the cell remained unchanged (Figure 4h). Similar to dsDNA, there were some cells in which the labeled FAM-dsRNA was “peeled off” from the surface during electrophoresis (Figure 4i). No variants where the pattern of FAM-dsRNA would be shifted along the cell periphery were revealed. These findings may imply that, in most cases, the primary interaction between dsRNA molecules (in contrast to that of dsDNA [15]) occurs with the rigidly anchored cell membrane elements and that dsRNA fragments are immediately internalized into the cell. Similar to dsDNA, we attribute the detected “peeling off” of the FAM-labeled material to interactions with the positively charged cell membrane elements. Meanwhile, there were no cell surface elements predetermining a strong physical contact with dsRNA fragments, which is potentially related to the G2/M phase of the cell cycle and the continuing cell division [16].

#### 2.4.3. The Effect of Blocking Specific Endocytosis Pathways on the Percentage of FAM+ Krebs-2 Cells

The analysis performed previously shows that FAM-dsRNA was internalized into Krebs-2 cancer cells. In order to identify the internalization mechanism, an extensive series of experiments was performed using compounds inhibiting different cellular pathways for the internalization of extracellular FAM-dsRNA, and the percentage of FAM+ cells was subsequently assessed. We analyzed the effect of nine inhibitors of different endocytosis pathways. To examine the endocytosis of FAM-dsRNA, we used dynasore as an inhibitor of dynamin-dependent endocytic pathways, chlorpromazine as an inhibitor of the clathrin-mediated endocytic pathways, and nystatin and methyl-β-cyclodextrin for inhibiting the lipid raft-mediated endocytic pathways, including the caveolin-mediated endocytic pathway. Cytochalasin D and EIPA were used as macropinocytosis inhibitors, wortmannin as an inhibitor of cellular phosphoinositide 3-kinases (inhibitor of receptor-mediated endocytosis), and sodium azide as an inhibitor of ATP synthesis.

Wortmannin 1.2 μg/mL and sodium azide 1.0 μg/mL were found to statistically significantly reduce the percentage of cells internalizing FAM-dsRNA (Figure 5). There was also a trend towards a decrease in the relative content of FAM+ cells when using nystatin. The abruptly increasing values at high inhibitor concentrations indicate that the membrane-dependent internalization has been completely inhibited (morphological analysis, flow cytometry).

#### 2.4.4. Final note on the internalization section

dsRNA internalization in CD34+ Krebs 2 stem cells can be considered as a general biological property of mouse CD34+ cells, including CD34+ hematopoietic progenitors. The following properties confirmate this assumption:(1)The localization of FAM labeled material in c-Kit+ (a marker of long-term hematopoietic stem cells) and CD34+ cells (a marker of short-term hematopoietic stem cells and multipotent progenitors);(2)The induction of the proliferation of primitive precursors treated with a dsRNA preparation;(3)Radioprotection and the formation of splenic colonies [9];(4)The rescue of hematopoietic lineage growth [10].

The above indicated that the dsRNA molecules were internalized in hematopoietic progenitors. dsRNA was immediately delivered to the internal compartments of the eukaryotic cell by clathrin-dependent endocytosis using the energy of ATP. Internalization was not affected by the cell charge and the presence of a heparin-binding domain. The data obtained indicate that the factors of internalization of dsDNA [15] and dsRNA into the cell are different.

### 2.5. Analysis of the Tumorigenic Potential of FAM+ Cells

An additional series of experiments were conducted to determine the tumorigenic potential of FAM+ cells. The findings obtained in our numerous earlier studies demonstrated that TAMRA+ Krebs-2 cells are cancer stem cells of Krebs-2 carcinoma. Since TAMRA+ cells also uptake FAM-dsRNA, it was natural to assume that FAM+ cells would also induce more rapid graft development compared with FAM− cells. Therefore, they would also have features of cancer stem cells. CBA mice were engrafted intramuscularly with 3 × 10^5^ FAM+ and FAM− cells each. The results were surprising; mice engrafted with FAM+ cells did not develop tumors. Mice engrafted with FAM− cells started to develop tumors on study day 2. In the control group, grafts started to be detected on day 6 post-grafting (Figure 6). Since the stem cell characteristics of TAMRA+ cells have been verified both in in vitro tests and by gene analysis [12,13,14], it is highly likely that internalized dsRNA molecules inhibit the processes ensuring proper functioning of Krebs-2 cancer stem cells.

### 2.6. Identifying the Critical Organ for Homing of FAM+ Hematopoietic Progenitors

At the initial stage of this phase of the study, it was shown that dsRNA degrades neither in mouse plasma (Figure 6C in [9]) nor directly in the bloodstream, where it is detected as native dsRNA for 3 h (Figure 7a). Therefore, one can expect that after the injection of dsRNA prior to γ-irradiation, dsRNA molecules will remain in the bloodstream and will be internalized into hematopoietic progenitors. After their mobilization, hematopoietic precursors rescued from radiation-induced destruction return to the bone marrow and are immobilized in the spleen to give rise to spleen colonies, as has been repeatedly shown in our experiments.

At this stage of the study, we attempted to answer whether lethally irradiated and mobilized hematopoietic progenitors can be immobilized in the splenic parenchyma and form spleen colonies, which is always observed for the respective treatments [9]. As shown in the previous sections, hematopoietic stem cells (c-Kit+) and poorly differentiated bone marrow cells (CD34+) internalize FAM-γP^32^-dsRNA; thus, having treated bone marrow cells with radioactively labeled dsRNA, one can observe homing of radioactively labeled progenitors. After being mobilized, hematopoietic precursors circulate in the bloodstream for about 2 h [17,18,19].

Bone marrow cells were isolated from mouse femoral bones, labeled with FAM-γP^32^-dsRNA, and washed to remove the nonincorporated material. After radioactivity measurements were performed, the cells were reinfused into recipient mice. Three hours later, the animals were euthanized; their organs were dissected, and radioactivity was measured using a RackBeta counter. The measurements were performed per gram of organ weight [20]. Two series of experiments were performed: one involved intact animals, and the other involved animals with severe leukopenia induced by the cytostatic agent cyclophosphamide [21]. Similar to γ radiation, cyclophosphamide depletes the hematopoietic organs. Thus, it is possible to identify the critical organs where hematopoietic progenitors will be immobilized using radioactively labeled hematopoietic precursors. It is known that cytoreduction is primarily observed in the bone marrow and spleen. Our findings demonstrated that the bone marrow and spleen were the critical organs for hematopoietic precursors circulating in the bloodstream in intact mice (Figure 7(b1)). When mice were exposed to cyclophosphamide, immobilized radioactively labeled cells were also detected in the lung parenchyma (Figure 7(b2)).

## 3. Discussion

Our pioneer study showed that dsRNA isolated from baker’s yeast and composed of many double-stranded ribosomal RNA fragments protected experimental animals against lethal doses of γ radiation [9]. That study also reported the cascade of events occurring when dsRNA enters the bloodstream, ensuring the protection of lethally irradiated mice: retention of dsRNA in the bloodstream, delivery of dsRNA fragments to hematopoietic precursors accompanied by escaping the lethal dose of γ radiation by hematopoietic progenitors, activation of the proliferation and mobilization of the rescued stem cells, homing to the splenic parenchyma, restoration of the immune and hematopoietic systems of irradiated mice, and the eventual survival of mice.

In the present study, we used synthetic dsRNA fragments. The structure of the artificial molecule included the two most frequently occurring dsRNA sequences in the original preparation [9]. This approach made it possible to partially preserve the structure corresponding to the original dsRNA and, at the same time, to analyze various biological parameters of dsRNA in a stable, unified form. We have modeled fragments of the entire process that had not been experimentally confirmed earlier and proved that (1) dsRNA is not degraded in mouse blood during the 2 h that it remains in peripheral blood, and (2) dsRNA is delivered to c-Kit/CD34+ hematopoietic progenitors. Extrapolation of the data obtained for Krebs-2 cancer stem cells suggests that dsRNA fragments are also delivered into hematopoietic progenitors in their native nondegraded form. (3) dsRNA affects the colony growth of bone marrow progenitor cells. Cells of the granulocyte–macrophage lineage exhibit the best response. Meanwhile, erythropoiesis is repressed. (4) The bone marrow and spleen are the critical organs responsible for homing FAM-γP^32^-dsRNA-labeled blood stem cells in the bloodstream. In other words, the concept of hematopoietic stem cell rescue and homing proposed by Ritter et al. [9] is valid and proven. Some critical reflections on the reported and discussed concept are presented below.

The relevant literature lacks the circulation data of RNA specific for dsRNA in peripheral blood. The fact that double-stranded RNA fragments are preserved for a long time ex vivo in plasma [9] and in vivo in the peripheral blood of mice agrees with the available literature data.

dsRNA internalization should be characterized separately. This study has shown that dsRNA is taken up by Krebs-2 cells. The FAM-labeled material is also detected in hematopoietic progenitors. A comparison of the earlier findings on radioprotection with the data obtained in the present study suggests that hematopoietic progenitors also internalize dsRNA.

Polyanion-binding receptors have been described and characterized fairly well [22,23,24]. They all, except for TLR3, have a cytoplasmic localization and cannot be responsible for the transition of dsRNA molecules across the cell membrane. We failed to find any reliable information in the open source literature indicating that TLR3 resides on the cell membrane of hematopoietic progenitors c-Kit and CD34 [25]. This means that there is another factor on the surface of hematopoietic progenitors responsible for binding and internalizing dsRNA. Moreover, the internalization mechanism should not contradict the characteristics of the revealed mode of dsRNA–cell interaction, namely, that (1) dsRNA binding to the cell is independent of cell charge, and (2) unlike dsDNA, dsRNA does not interact with heparin-binding sites of proteoglycans/glycoproteins, scavenger receptors, and glycosylphosphatidylinositol-associated proteins. One of the details of this difference in the interaction between the two types of polyanions has been illustrated well by X-ray crystallographic analysis in Garlatti et al. [26]. The interaction between dsRNA and the cell is accompanied by dsRNA internalization and immobilization inside the cell. In some circumstances, dsRNA (in a manner similar to dsDNA) can interact with the cell without any subsequent internalization, as confirmed by the image in Figure 4i demonstrating that the FAM material is peeled off from the cell surface.

The experiments with inhibitors of different endocytosis pathways indicate that the mechanism of dsRNA internalization is related to the receptor-mediated process that requires energy from ATP. If we do not take into account the glycocalyx components, scavenger receptors (probably the glycosylphosphatidylinositol-anchored proteins) can potentially act as dsRNA-binding proteins. Several types of scavenger receptors (namely, SR-AI, MARCO, SCARB1, SCARF1, and STAB2) are known to bind polyanions, including dsRNA [27,28,29]. These very types of receptors can be responsible for the uptake and internalization of dsRNA fragments into cells belonging to the population of poorly differentiated cells of different lineages.

It has been hypothesized that dsRNA fragments taken up by c-Kit/CD34 hematopoietic progenitors in irradiated mice can be involved in several processes. First, they differentially affect the colony formation of cells of the erythroid and granulocyte–macrophage lineages (Figure 1b, Table 1). Second, they participate in the repair of γ-radiation-induced damage.

Most poorly differentiated hematopoietic stem cells are believed to exist in the dormant G0 state. Nevertheless, some hematopoietic progenitors in the bone marrow undergo cell division; moreover, differentiation direction upon treatment depends on the cell cycle stage [30]. This observation means that the data obtained by Ritter et al. [9] suggesting dsRNA is involved in the repair of double-strand breaks induced by interchain cross-links generated in dividing cells by γ irradiation are correct and confirm the cell cycle status of these progenitors. The findings reported in other studies indicate that dsRNA can be involved in damage repair as an external matrix. This involvement restores the integrity of the chromatin continuum damaged by radiation [31,32].

Along with the expected rescue of hematopoietic progenitors initiated by the interference of the double-strand break repair occurring in dividing progenitor cells, there may be other alternatives for viability preservation in hematopoietic progenitors. dsRNA may be involved in mitigating the stress state reported for angiogenin [33,34,35,36,37]. Furthermore, dsRNA taken up by stem cells may trigger the synthesis and secretion of IL-1 and IL-8, which are involved in the post-irradiation rescue of stem cells [38,39,40,41,42,43,44,45]. dsRNA can also trigger antioxidant responses associated with the thrombospondin system [46,47,48,49,50]. Although several potential variants of rescue mechanisms are listed above, it is fair to say that the rescue of c-Kit/CD34 cells due to their internalization of extracellular dsRNA fragments is a prerequisite for the radioprotective effect of dsRNA.

The two results obtained in this study create a bridge between the two states of hematopoietic progenitors before their contact with dsRNA and after dsRNA internalization into stem cells and γ irradiation.

As follows from the colony stimulation experiments, dsRNA stimulates the differentiation of granulocyte–macrophage progenitors. Meanwhile, our findings suggest that intracellular dsRNA will serve as a prerequisite for maintaining the viability of cells of the granulocyte–macrophage lineage that have started to proliferate.

It was found, simultaneously, that bone marrow stem cells radioactively labeled with γP^32^-dsRNA that were reinfused into the bloodstream are immobilized in the spleen and bone marrow (in some cases, mobilized stem cells can also reside in the lungs [51]).

Together, these findings suggest that, in the irradiation experiments after dsRNA internalization, the proliferating rescued granulocyte–macrophage stem cells go into the bloodstream and are immobilized in the spleen. This organ is a place where the proliferative potential of stimulated progenitors is implemented. It is generally believed that stem cell homing is an inherent biological property of hematopoietic progenitors. The bone marrow and spleen are the main compartments where mobilized circulating hematopoietic stem cells are anchored. In the spleen, the mobilized progenitors give rise to clearly discernible colonies derived from a single hematopoietic progenitor and consist of a mixture of mature cells of the myeloid lineage [51,52]. This means that the data obtained in this study, and earlier [15], are consistent with the biological characteristics of hematopoietic stem cells.

The cytologic evaluation has revealed structures where FAM+ cells reside in the center of cellular structures formed by five to nine cells (Figure 1a). This conformation of tightly bound hematopoietic stem cells can represent the arrangement of cells that forms the hematopoietic stem cell niche [53,54,55,56,57].

All the available data characterizing the stem cell properties of FAM+ cells both in the bone marrow and Krebs-2 ascites were compared. Thus, FAM-dsRNA was found to internalize c-Kit/CD34+ cells, which are long-term or short-term hematopoietic stem cells or multipotent progenitors [11,58,59,60,61,62]. It was revealed, simultaneously, that dsRNA stimulates colony formation by bone marrow cells, thus indicating that it affects early bone marrow progenitors. FAM+ bone marrow and Krebs-2 cells were double-positive for CD34, a marker of hematopoietic progenitors. It was shown that FAM+ Krebs-2 cells can simultaneously uptake TAMRA-dsDNA, while TAMRA+ cells are Krebs-2 cancer stem cells [12]. Together, these results indicate that FAM+ cells are bone marrow stem cells whose functional activity regulates the survival of lethally irradiated mice, and the images in Figure 1a show the cellular components of bone marrow stem cell niches.

The results of the experiments involving the transplantation of FAM+ Krebs-2 cells were surprising and needed to be studied further. Grafting of FAM– cells resulted in tumor development 12 days post-transplantation, and graft failure occurred with FAM+ cells. We interpret these results as showing that graft development in a negative population of transplanted cells (FAM−) is associated with the presence of a certain number of stem cells (~20%) undergoing the G2/M phase when all the metabolic processes are completely stopped, and the reconstruction of the actin cytoskeleton takes place. In this cell-cycle phase, stem cells do not internalize dsRNA. After the transition to the G1 phase, they yield later tumor development compared with the control. The lack of grafting in the group with re-transplanted FAM+ cells can be explained as follows. The synthetic dsRNA is composed of two portions, being the domains of ribosomal RNA. It is assumed that once dsRNA enters the nucleus of cancer stem cells, it induces RNA interference and inhibits cellular synthesis. We have previously shown that linear dsDNA internalized into the cell forms a ring-shaped structure after processing [63,64]. It is well known that double-stranded DNA ends act as an extremely strong molecular irritant to the cell and trigger repair, resulting in the removal of double-stranded DNA ends from the intracellular space. Our study showed that dsRNA is neither processed nor ligated in any form inside the cell. In other words, dsRNA does not activate the repair mechanisms for removing free double-stranded RNA ends and can remain inside the nucleus for a long time in its native form. This property of dsRNA molecules delivered into Krebs-2 stem cells is apparently a condition for activating the interference mechanism and inhibiting the synthesis in cancer stem cells, which was responsible for the absence of graft growth.

## 4. Materials and Methods

### 4.1. Experimental Animals and Cells

We used 2- to 6-month-old CBA/Lac mice (males and females, 18–24 g) bred at the Common Use Center Vivarium for Conventional Animals, Institute of Cytology and Genetics of the Siberian Branch of the Russian Academy of Sciences (Novosibirsk, Russian Federation). Animals were grown in groups (6–10 mice per cage) with free access to food and water.

All the animal experiments were performed in accordance with the European Convention for the Protection of Vertebrate Animals used for Experimental and other Scientific Purposes and the protocol approved by the Interinstitutional Bioethics Commission of the Institute of Cytology and Genetics of the Siberian Branch of the Russian Academy of Sciences (Protocol N 48/4 from 18.03.2019). Animals were sacrificed using the method of cervical dislocation.

To obtain murine bone marrow cells, the bone marrow mass was washed out of the tubular bones of intact mice with phosphate-buffered saline (PBS) and thoroughly resuspended. Cells were washed twice and counted.

Krebs-2 ascitic cells were obtained from mice with a Krebs-2 tumor. Krebs-2 ascitic carcinoma is a strain-nonspecific tumor derived from epithelial cells. This cancer cell line was obtained from the cell depository of the Institute of Cytology and Genetics (Novosibirsk, Russia) and is maintained in mice as a transplanted tumor. To obtain ascites-bearing mice, mice were intraperitoneally inoculated with 2 × 10^6^ Krebs-2 cells. On day 7 after inoculation, a sample of Krebs-2 ascitic cells can be taken.

### 4.2. Nucleic Acid Probes

The 68 bp dsRNA labeled with 6-carboxyfluorescein (6-FAM) at their 3′ ends were synthesized using the procedure described in Ritter et al. [9] and designated as FAM-dsRNA.

All the procedures involving carboxytetramethylrhodamine (TAMRA)-labeled dsDNA probe preparation were performed as described in Dolgova et al. [12] and designated as TAMRA-dsDNA.

dsRNA (1 µg) was labeled with γP^32^ ATP using T4 polynucleotide kinase at 37 °C for 30 min. The reaction was stopped by adding EDTA to a concentration of 20 mM; RNA was purified by extraction with phenol-chloroform and precipitated using 3 volumes of ethanol.

### 4.3. Internalization of Labeled Nucleic Acids into the Bone Marrow Cells or Krebs-2 Cells

Murine bone marrow cells or Krebs-2 ascitic cells (0.5 × 10^6^) were incubated in the presence of 0.1 µg of FAM-dsRNA or TAMRA-dsDNA at room temperature in the dark for 30 min.

In some experiments, after the incubation with labeled probes, cells were stained with PE-labeled anti-CD34 antibodies (BD Biosciences, San Jose, CA, USA) or FITC-labeled anti-CD34 antibodies (Sony Biotechnology, San Jose, CA, USA) according to the manufacturer’s protocol in the dark at room temperature for 30 min.

In some experiments, before or after incubation, cells were stained with 2.7 µg/mL Basic Blue 41 (BB41) (Merck, Darmstadt, Germany) at room temperature for 20 min. After incubation, cells were washed with PBS twice and deposited onto glass slides using the cytospin technique. The deposited cells were treated with DAPI/DABCO/glycerol and assayed using an Axio Imager M1 microscope (Zeiss, Oberkochen, Germany). In several cases, live cells were analyzed on an LSM 510 META microscope (Zeiss, Germany). Microscopic assays were performed at the Common Use Center for Microscopy of Biologic Objects of the Siberian Branch of the Russian Academy of Sciences.

FACS analysis of cells was carried out using a BD FACSAria™ III flow cytometer at the Center for the Collective Use of Flow Cytometry of the Institute of Cytology and Genetics of the Siberian Branch of the Russian Academy of Sciences.

### 4.4. Stimulation of Bone Marrow Colony Growth by dsRNA

For activation, 1 × 10^6^ bone marrow cells were incubated with 0.05–0.25 µg dsRNA in IMDM supplemented with 2% FBS for 2–2.5 h in the presence of 5% CO_2_ at 95% humidity and 37 °C. For myeloid progenitor quantification, 2 × 10^4^ whole bone marrow cells were plated in MethoCult M3434 methylcellulose (Stem Cell Technologies, Vancouver, BC, Canada) according to the manufacturer’s instructions. Colonies were scored by visualization on days 7–10.

### 4.5. Autoradiographic Assay

Cells treated with FAM-γP^32^-dsRNA were sorted by the FAM+/FAM– parameter. The sorted cells were washed with normal saline twice, and RNA was extracted using TRIzol (Life Technologies, Wilmington, DE, USA). RNA extracted from the cells was applied to 1% agarose gel, and electrophoresis was performed. The gel was dried, X-ray film was placed over it, and it was exposed. The radiograph was analyzed in transmitted light.

### 4.6. Microgel Electrophoresis and Membrane Electrophoresis of Cells

The cells (0.2 × 10^6^ cells) were deposited onto a glass slide using the cytospin technique; 20 µL of 1% low-melting point agarose with DAPI-DAPCO was placed over it, and the glass slide was covered with a cover slip. Alternatively, 1 × 10^6^ cells in 1 mL of normal saline were placed into the dialysis membrane.

The specimen or the tightly closed membrane was placed into an electrophoresis chamber; its contact with normal saline used as an electrophoresis buffer was ensured using 3 MM filter paper. Electrophoresis was carried out at 4 V/cm for 15–30 min.

After electrophoresis, the specimens were immediately analyzed microscopically or prepared from the cells collected from two halves (from the sides of the positively and negatively charged electrodes) after squeezing the membrane along its midline.

### 4.7. Treating Cells with Endocytosis Inhibitors

The cells were incubated with inhibitors (heparin, sodium azide, EIPA, cytochalasin D, methyl-β-cyclodextrin, nystatin, dynasore, wortmannin, chlorpromazine, and dextran) at concentrations displayed in the figures at 37 °C for 30 min. The cells were then washed and treated with FAM-dsRNA.

### 4.8. Grafting Cells to Mice

Krebs-2 cells treated with FAM-dsRNA were sorted by the FAM+/FAM– parameter. Mice (*n* = 3 in each group) were intramuscularly grafted with 0.3 × 10^6^ unsorted Krebs-2 cells (control), FAM+, or FAM– Krebs-2 cells in the hind paw.

### 4.9. Analyzing the Stability of γP^32^-dsRNA in Mouse Bloodstream

The mice received an intravenous injection of 2 µg γP^32^-dsRNA. After 5, 60, and 120 min, blood samples were collected from the caudal vein. Blood samples of equal volume were applied onto an agarose gel, and electrophoresis was performed. The gel was dried, X-ray film was placed over it, and it was exposed. The resulting radiograph was analyzed in transmitted light.

### 4.10. Analysis of the Amount of Radioactive Label in Organs of Mice after Intravenous Injection of fam-γp^32^-dsrna-Labeled Bone Marrow Cells

Mice received an intravenous injection of FAM-γP^32^-dsRNA-labeled bone marrow cells (*n* = 4 in each group). Three days before, leukopenia was induced in some mice by injecting 200 mg/kg cyclophosphamide. Intermediate blood samples were collected from the caudal vein. Three hours after injection, mice were euthanized by cervical dislocation. Organs were isolated and transferred into special vials for dosimetry experiments. Dosimetry was conducted using a RackBeta liquid scintillation counter. The values were normalized to the weight of the analyzed organ.

## 5. Conclusions

The experiments conducted in this study directly demonstrate that synthetic double-stranded RNA is internalized into a eukaryotic cell via a natural mechanism. After interacting with hematopoietic progenitors, the labeled material is detected within their internal compartments. This interaction also initiates the proliferation of these cells. Mobilized hematopoietic progenitors repopulate the spleen and bone marrow. As a result, treated animals survive the irradiation with a lethal dose of 9.4 Gy.

Further research is needed into the factors and mechanisms involved in the internalization of dsRNA, a novel property of poorly differentiated cells yet to be described.

## Figures and Tables

**Figure 1 ijms-24-04858-f001:**
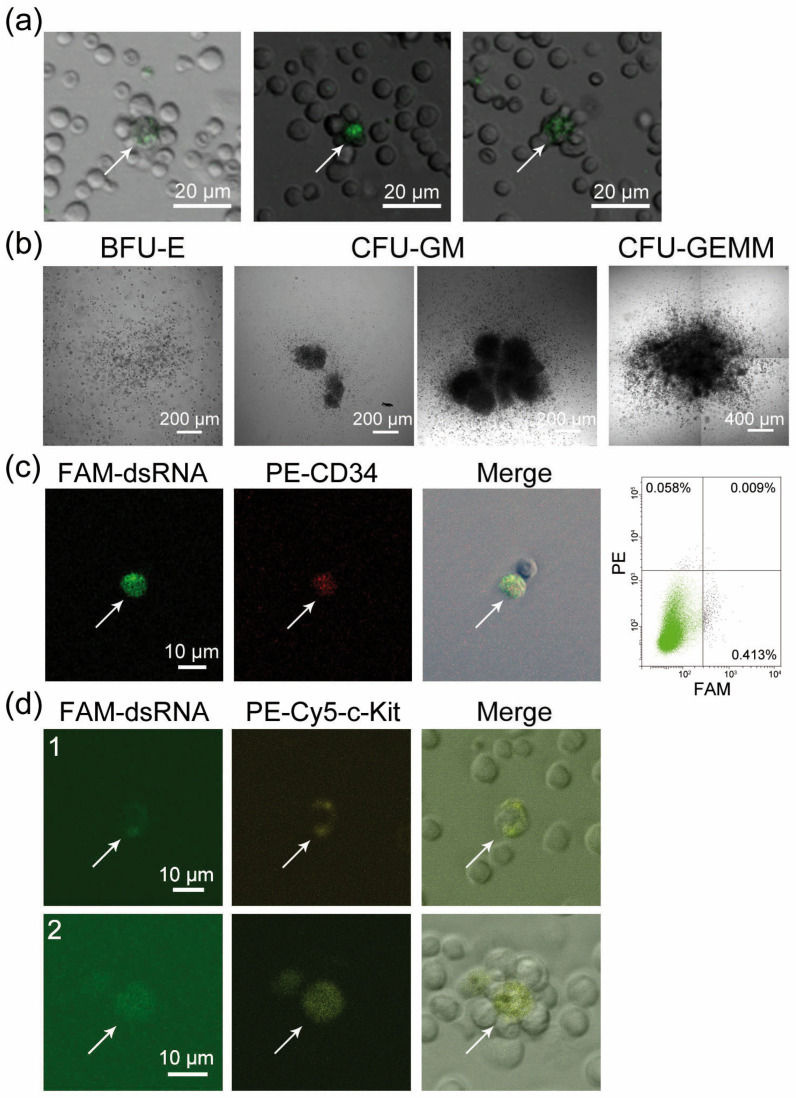
Cytologic evaluation of bone marrow cells as a target of dsRNA action. (**a**) A panel of rosettes, presumably the bone marrow stem cell niches, after incubating cells in the presence of FAM-dsRNA. A FAM+ cell resides in the rosette center. (**b**) Morphology of the colonies formed by bone marrow hematopoietic progenitors after stimulation by synthetic dsRNA. (**c**) The CD34+/FAM+ bone marrow cell double-positive for two markers. Bone marrow cells were treated with FAM-dsRNA and PE-conjugated anti-CD34 antibodies; CD34+ cells were sorted and analyzed by confocal microscopy. A flow cytometry data plot demonstrating the quantitative content of cells in the bone marrow specimen is shown on the right-hand side inset. (**d**) Bone marrow cells double-positive for FAM (green) and PE-Cy5-conjugated c-Kit marker (yellow). 1—an individual cell; 2—a cell within a rosette, presumably the bone marrow stem cell niche. Arrows indicate stained cells.

**Figure 2 ijms-24-04858-f002:**
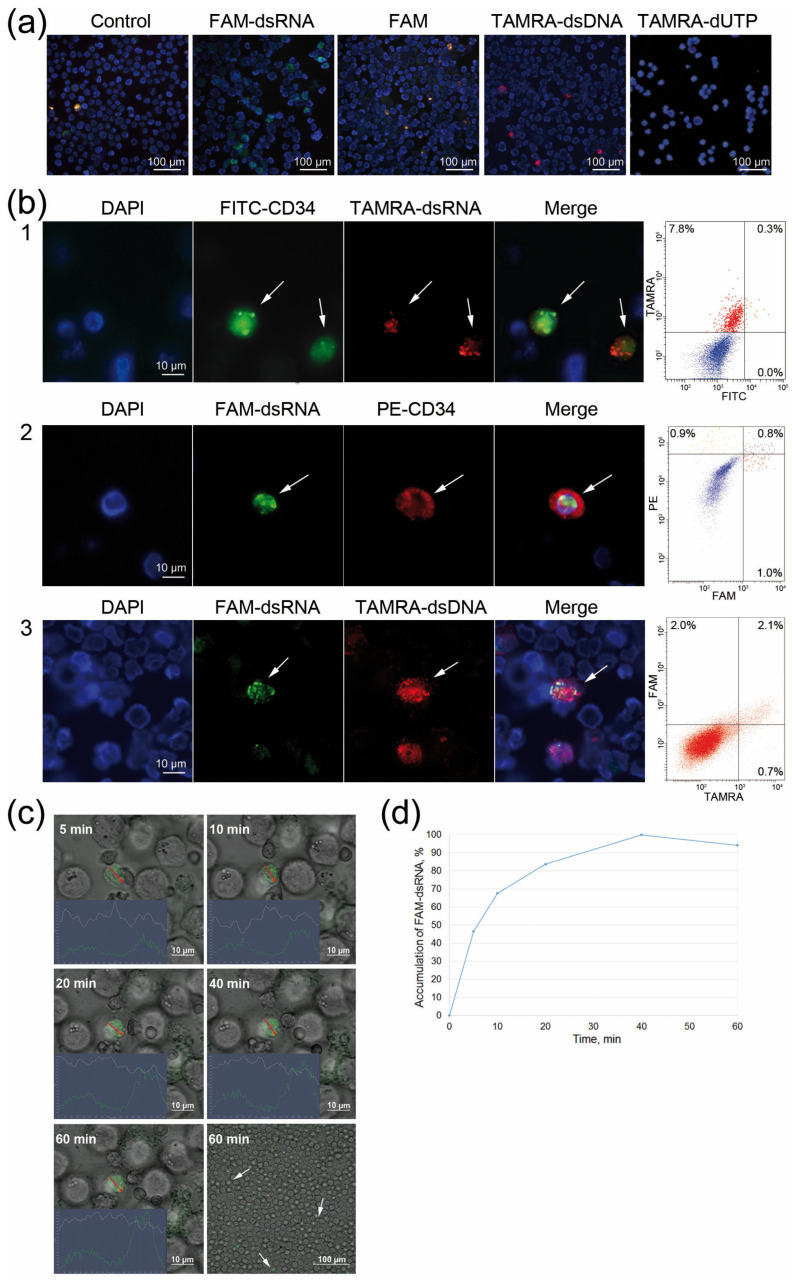
Cytologic evaluation of internalization of FAM-dsRNA and TAMRA-dsDNA fragments into CD34+ Krebs-2 cells. (**a**) Krebs-2 cells treated with FAM, TAMRA-dUTP, FAM-dsRNA, and TAMRA-dsDNA. (**b**) Krebs-2 cells treated with: (1) FITC-labeled anti-CD34 antibodies (green) and TAMRA-dsRNA (red), (2) FAM-dsRNA (green) and PE-labeled anti-CD34 antibodies (red), and (3) FAM-dsRNA (green) and TAMRA-dsDNA (red). Arrows indicate stained cells. Flow cytometry data for the treated cell specimens are shown on the right-hand-side inset. (**c**) Microscopy analysis characterizing the dynamics of FAM-dsRNA accumulation in the same Krebs-2 cell. The images were recorded 5, 10, 20, 40, and 60 min after the labeled probe was added to the medium. The plots in the left corner of the images characterize the FAM intensity along the analyzed line (shown with a red arrow). (**d**) The diagram showing the dynamics of FAM-dsRNA accumulation in Krebs-2 cells.

**Figure 3 ijms-24-04858-f003:**
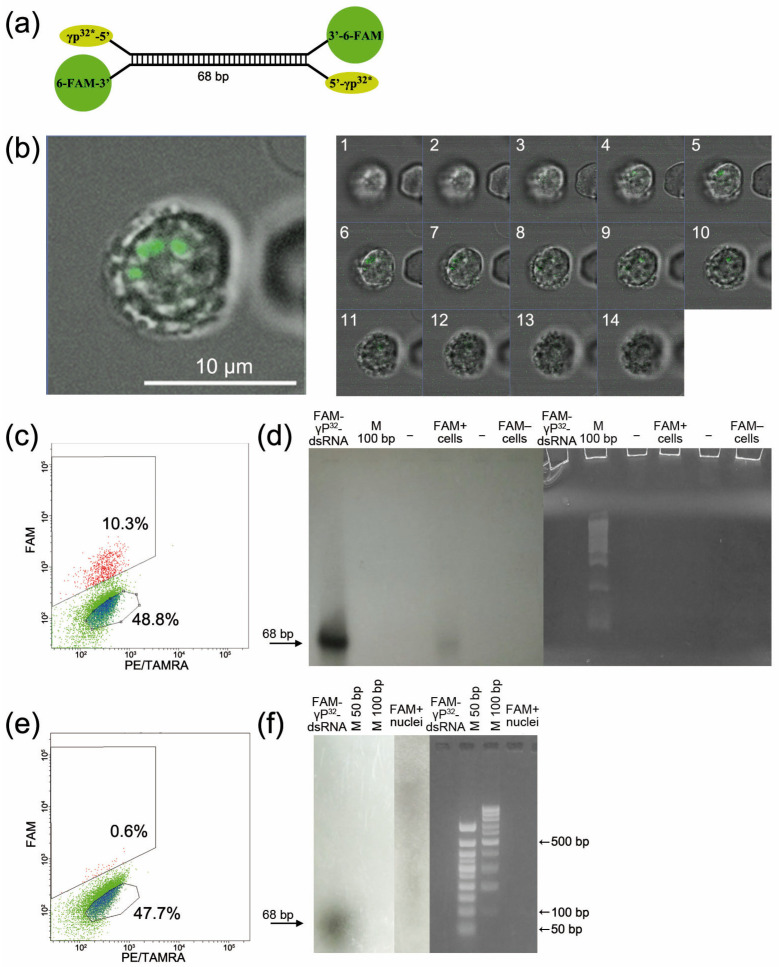
Internalization of dsRNA into Krebs-2 cells. (**a**) Design of inserting labeled adducts into the synthetic dsRNA. (**b**) Confocal analysis data for FAM-dsRNA internalization into a Krebs-2 cell. One cell and its slices in different planes (1–14) are shown. (**c**) The plot demonstrating the cell population used for the assay. (**d**) Polyacrylamide gel and its autoradiogram demonstrating that labeled dsRNA is present in FAM+ cells and absent in FAM− cells. (**e**) The plot demonstrating the cell population used for analyzing the nuclei. (**f**) Electrophoresis gel (2% agarose) and its autoradiogram demonstrating that labeled dsRNA is present in the nuclei of FAM+ cells.

**Figure 4 ijms-24-04858-f004:**
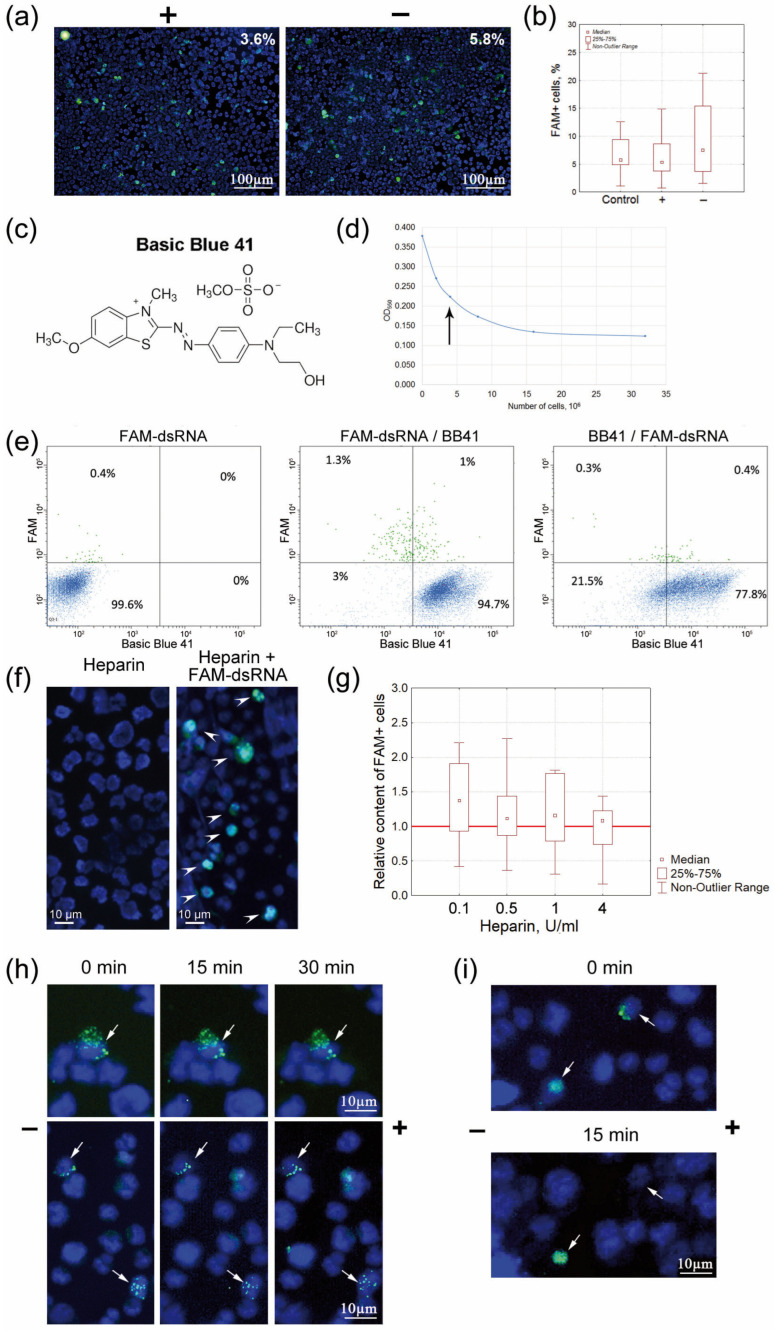
Characterization of selected cell surface parameters potentially related to the internalization of FAM-dsRNA into Krebs-2 cells. (**a**) The microscopic images of Krebs-2 cells isolated from two halves of a mini electrophoresis chamber and treated with FAM-dsRNA. The percentage of cells that internalized the labeled probe is shown in the upper right-hand corner. (**b**) The cumulative plot showing the number of FAM+ cells at both poles of the electrophoresis chamber built using the data from five independent experiments. (**c**) A Basic Blue 41 molecule. (**d**) The curve showing the saturation of Krebs-2 cells with the dye Basic Blue 41. (**e**) Flow cytometry analysis of the internalization of the sequentially added FAM-dsRNA and Basic Blue 41; Basic Blue 41 and FAM-dsRNA by Krebs-2 cells. The first block shows the control distribution of cells treated with FAM-dsRNA. (**f**) The microimages of Krebs-2 cells after the addition of heparin and incubation with FAM-dsRNA. The arrows denote the FAM-labeled material. (**g**) Effect of heparin on the internalization of FAM-dsRNA into Krebs-2 cells. The presented plot shows the contents of FAM+ cells after adding different amounts of heparin (*n* = 10). The red line denotes the content of FAM+ cells among Krebs-2 cells in the absence of heparin set equal to 1. (**h**,**i**) Microgel electrophoresis. Krebs-2 cells after treatment with FAM-dsRNA, cytospinning, embedding into 1% low-melting agarose, and electrophoresis on a microscope glass slide. Electrophoresis duration is shown above the blocks. The “−” and “+” are poles of electrophoretic separation. The arrows denote the FAM-labeled material. (**h**) FAM-dsRNA does not change its spatial position during electrophoretic separation. The duration of incubation in the presence of the probe is 40 min. (**i**) FAM-dsRNA is “peeled off” during electrophoresis. A cell from the same electric field where the labeled material does not change its position is shown for clarity. The duration of incubation in the presence of the probe is 60 min.

**Figure 5 ijms-24-04858-f005:**
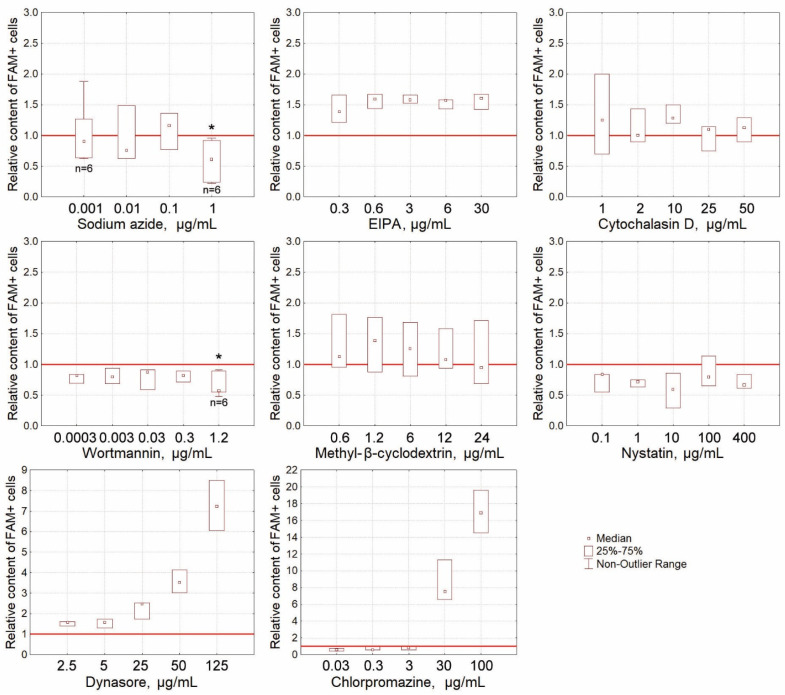
Inhibition of internalization of FAM-dsRNA into Krebs-2 cells by various agents. The presented plots show the contents of FAM+ cells during inhibition of internalization into the cell using different amounts of respective inhibitors. The asterisks (*) denote the values being statistically different from the control (no inhibitor used) (*p* < 0.05; the Wilcoxon matched pairs test). The red line denotes the content of FAM+ cells among Krebs-2 cells in the absence of inhibitors set equal to 1 (*n* = 3 in all cases if not otherwise specially indicated).

**Figure 6 ijms-24-04858-f006:**
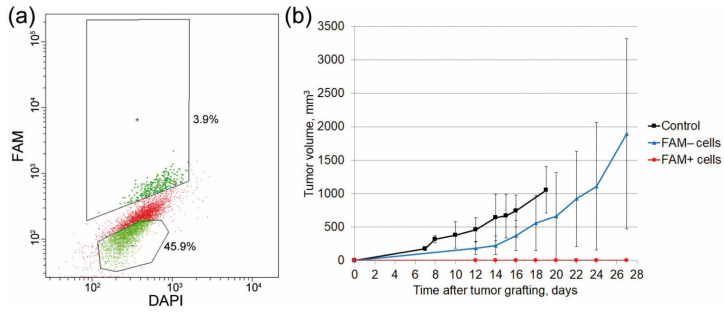
Graft development in mice with re-transplanted FAM+ and FAM− grafts. (**a**) The flow cytometry plot showing the distribution of re-transplanted populations of Krebs-2 cells. (**b**) The dynamics of tumor growth in mice with re-transplanted FAM+ and FAM− grafts compared with the control group mice engrafted with Krebs-2 cells. Means ± standard deviations are presented.

**Figure 7 ijms-24-04858-f007:**
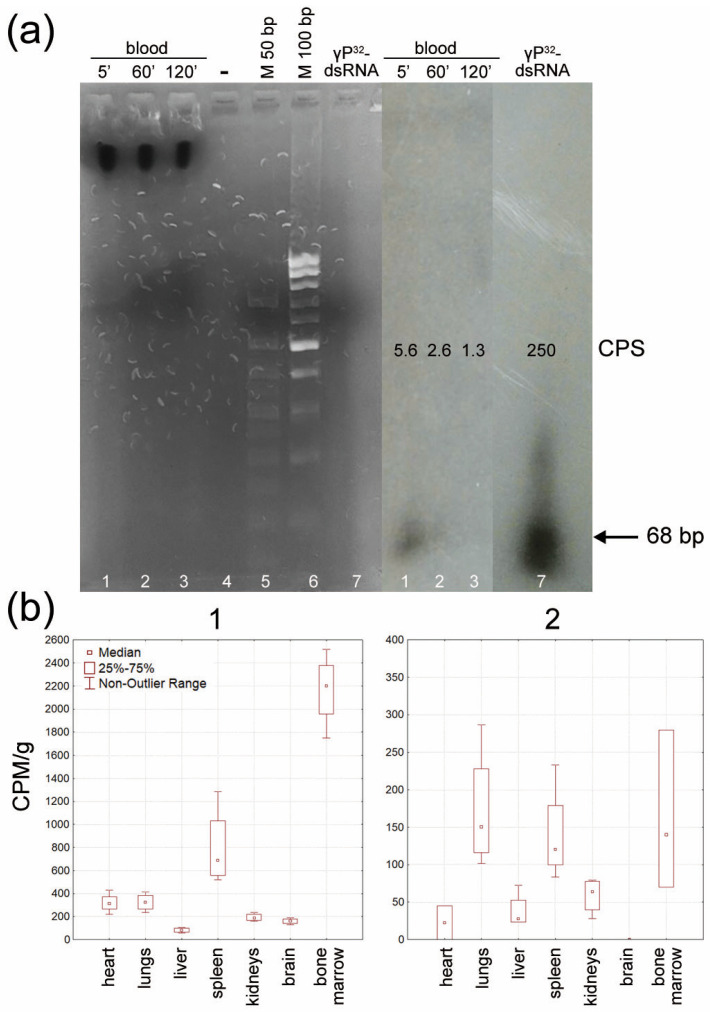
Identifying the critical organ for homing of FAM+ hematopoietic progenitors. (**a**) Conservation of linear dimensions of dsRNA in the bloodstream of an intact animal. (**b**) Homing of bone marrow cells radioactively labeled with FAM-γP^32^-dsRNA in intact mice (1) and mice with cyclophosphamide-induced leukopenia (2).

**Table 1 ijms-24-04858-t001:** The number of colonies in accordance with their hematopoietic lineage. The mean ± standard deviation is presented.

Colony Type	Control	dsRNA
BFU-E	13.5 ± 0.7	9.5 ± 0.7
CFU-GM	33.5 ± 6.4	45.0 ± 0.1
CFU-GEMM	2.5 ± 0.7	2.0 ± 1.4

## Data Availability

The data presented in this study are available within the article.

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
