# Peer review of "Impact of Double-Stranded RNA Internalization on Hematopoietic Progenitors and Krebs-2 Cells and Mechanism"

_ijms, 2023, doi:10.3390/ijms24054858_

Round 1

Reviewer 1 Report

In the manuscript entitled "Internalization of double-stranded RNA into hematopoietic progenitors and Krebs-2 cancer stem cells", the 68 bp synthetic dsRNA was found to be uptaken by hematopoietic progenitors and Krebs-2 cancer stem cells. The research is interesting, however, several concerns should be addressed before further processing of the manuscript.

- For most of the flowcytometry graphs, controls are missing.

- The sources of the cells used in experiments should be provided.

- In Fig 1d, FAM-dsRNA and PE-Cy5-conjugated c-Kit marker presented similar colors, the merged images seem difficult to distinguish.

- In Line 2, Fig 2b, the merged image is not consistent with the photograph of a single channel.

- In Fig 2c,d, if only one cell is analyzed, the data is not solid enough. 

Reviewer 2 Report

Ritter et al., in their report, demonstrated that dsRNA is directly delivered into HPC cells that induce its proliferation. Furthermore, they describe the possible mechanism and stability of the dsRNA in vivo.

There is a lot of data in the study however, the study is hard to read due to the English usage.

I strongly recommend the manuscript should be thoroughly reviewed by a non-expert native English speaker before the further review process.

In some figures, the number of positive cells is not enough to derive a meaningful conclusion. Some important controls are also missing.

Figure 1C: Please mark dead cells in the flow-derived graph.

Figure 2 :b: Surprisingly there were no cells that were dsRNA positive and CD34. The percentage of positive cells is low to derive a fruitful conclusion.

Figure 3F: Why FAM+nuclei is taken from a different set of experiment?

Were different sequences of dsRNA used in any study? If yes what was the result?

Did the authors observe any dsRNA-related cell toxicity?

Please reframe lines 33-34.

Please replace ref 1 with the original studies.

Please reframe lines 51-52. The meaning is not clear.

Please do not start a sentence with a number. For example lines 75 &78.

Add a reference to lines 75, and 78. 80 and 84.

Line 100: reframe.

Line 114-115: Please reframe.

The topic of the study should be changed. What was the impact of the internalization of dsRNA into HPC and Krebs-2 cells that should be incorporated in the title?

Reviewer 3 Report

The study presented by Ritter and colleagues is very interesting and well-conceived. There are no severe flaws in the experimental design, however, there are some issues that the authors have to clarify before publication:

1) In the following sentence, why did you focus the attention on γ rays only? “Poorly differentiated bone marrow cells, testicular germ cells, as well as the intestinal and skin epithelium are the main target organs for γ rays [2,3].”. Indeed, the major source of radiation exposure is due to other type of ray (e.g. UV-A, UV-B and UV-C rays). Please, clarify;

2) In the Introduction section, the authors have to briefly describe the main sources of γ-rays;

3) I suggest to provide the following data in a narrative way. In addition, you have to provide references supporting all these data:         

• 160–200 µg dsRNA per mouse injected one hour prior to exposure to radiation pro- 75 tects the experimental animals against the lethal radiation dose. The radioprotective effi- 76 cacy is identical or higher than that of the conventional radioprotector B-190 (Russia).                                                                                    

• 160–200 µg dsRNA per mouse injected on day 4 post-irradiation prevents death of  60% experimental mice.

• dsRNA is not degraded in mouse plasma; its size remains unchanged after 2 hr in-cubation

• the double-stranded form of RNA rather than its sequence is important for the radioprotective activity.

• FAM-dsRNA is detected in CD34+ hematopoietic multipotent progenitors after 1 h incubation of bone marrow cells in the presence of dsRNA.

• the protective effect does not correlate with repair proceeding via the non-homolo- 86 gous end-joining mechanism but correlates with repair occurring via the homologous re- 87 combination mechanism induced by radiation-triggered damage in the cells.

• survival of experimental mice correlates with emergence of a large number of proliferating leukocytes in the spleen that form typical lymphoid cell colonies.

4) The authors presented the electrophoresis data as a collage of different images. This is confusing and graphically wrong. They should also provide the original gel images as supplementary materials with a clear description of each experiment;

5) Did the authors collect data on the DNA damages observed in cells exposed to γ-rays treated or not with dsRNA. This is a key point of the manuscript;

6) Overall, it is not clear what is the scientific question of the study;

7) In the following sentence please clarify what kind of cells, the name, the sources (commercial or patient-derived) and all relevant information: “Cells (0.5×106) were incubated in the presence of 0.1 µg of FAM-dsRNA or TAMRA-dsDNA at room temperature in the dark for 30 min.”.

Round 2

Reviewer 1 Report

The manuscript has been improved.

Reviewer 2 Report

The team has vigorously worked on the manuscript. The manuscript looks much improved upon revision. 

Most of the queries have been promptly rectified. The manuscript may be accepted for publication after correcting minor typographical errors.

I congratulate the team on their excellent work.